# Talent Management in Healthcare: A Systematic Qualitative Review

Konstantinos D. Mitosis [1], Demetris Lamnisos [2] and Michael A. Talias [1,*]

1    Healthcare Management Postgraduate Program, Open University Cyprus, P.O. Box 12794, Nicosia 2252, Cyprus; konstantinos.mitosis1@st.ouc.ac.cy

2    Department of Health Sciences, School of Science, European University of Cyprus, Nicosia 2404, Cyprus; D.Lamnisos@euc.ac.cy

*    Correspondence: michael.talias@ouc.ac.cy

**Abstract:** Talent Management (T.M.) constitutes a modern and emerging research area in Human Resources Management (HRM). Using a systematic literature approach, we searched in Talent Management literature in the healthcare sector context. We conclude that the number of related studies is minimal. The benefits of implementing Talent Management strategies in healthcare organizations are essential for the organization's sustainable development and the talented staff and healthcare services patients. Our goal is to undertake a systematic literature review to identify these factors related to talent management practices suitable for healthcare organizations and professionals. We have conducted, according to PRISMA guidelines, a systematic literature review (2010–2020) in the electronic databases PubMed, CINAHL, Cochrane Database of Systematic Reviews, Health Source/Nursing Academic Edition. Search terms related to T.M. were ("Talent Management" AND "Talent Healthcare"). Strict inclusion and exclusion criteria were set for observational studies, while grey and unpublished literature, uncontrolled studies, protocols, commentaries, and conference proceedings were excluded. All included items were assessed for their quality according to set criteria. Six hundred and eighty-four studies were identified, of which 24 met the requirements. The resulting Talent Management Factors were grouped into nine categories: Programming, Attraction, Development, Preservation, Performance Assessment, Work Climate, Culture, Succession Planning, and Leadership. Based on these factors, we provide a holistic picture of the referred domain's leading developments. The paper determines the Talent Management factors and explains what happens in practice. In this way, we contribute to building a theoretical framework for T.M. in terms of the organizational context.

**Keywords:** talent management; healthcare; systematic literature review; qualitative content analysis; strategy; leadership; quality services; sustainable services; organizational commitment

## 1. Introduction

Most theorists on Management describe Talent Management (T.M.) as the process of attraction, development, and preservation of working people of increased abilities, skills, and knowledge [1,2] T.M. constitutes a modern research field in Human Resources Management (HRM) and is vital to survival and the organizations' competitive advantage. Through HRM, organizations manage their staff in the best possible way to increase productivity, effectiveness, and quality of their services [3–5]. Implementing T.M. strategies for the health services organizations positively impacts the organization itself, the personnel, and the patients-beneficiaries.

There are several research studies in the international bibliography supporting the positive effect of T.M. on outflows in the field of health [6–8]. A recent overview of T.M. in nursing concludes that the implementation of T.M. strategies enhances nurses' clinical skills, increases the personnel's work satisfaction, improves specialized medical attendants' skills, and increases the organization's efficiency therapy rates of the patients [9]. Some

studies support the positive correlation between talent management strategies in healthcare units and employees and organizations [10,11]. Health-services-providing organizations constitute perhaps the most complex service-providing organizations [12,13]. These employ many specialists (doctors, nurses, administrators, paramedics, pharmacists, assistants, and others) who cooperate directly to provide services. Health as a "good" constitutes for humans what is most valuable that they possess; therefore, the pressure for more quality outflows (patient therapy) from the health-services-providing units are ever-increasing [14].

Health today is also confronted with modern challenges such as health globalization and internationalization, reduction of disposable funds, lack of a talented personnel pool, population aging, and technology development. Also, at the global level, the migration of high-performance healthcare professionals and workers from different healthcare sectors between countries is intense and facilitated by the "international recruitment industry" [15]. According to the latest research, shortages in the health sector workforce are likely to exceed 15 million workers soon [16]. The Association of American Medical Colleges argued that the United States would face a shortage of physicians over the next decade, highlighting the American Medical Colleges [17]. Addressing the lack of high-performance staff reinforces the need to adopt talent management strategies and integrated talent management systems from healthcare providers to address the sector's current challenges. At the same time, international literature argues that implementing integrated Talent Management systems would improve organizations' efficiency, reduce production costs, and reduce patient health risk [18–20]. In such a complex environment as the afore-described, the attraction, development, and preservation of talented personnel prove to be challenging, composite, and rigorous tasks [21].

The difficulties in introducing T.M. strategies focus on the lack of a clear definition, the lack of competent staff to promote the implementation of T.M. systems, the prior knowledge of what works and stressed the urgent need to ensure reliable research on the subject as well as the assessment of the performance of T.M. in health care [22].

It has been widely accepted in the scientific society that the only alternative to achieving the ever-demanding goals of health-services-providing organizations, namely, providing quality services and competitive advantage, is implementing HRM and T.M. methods [23–25]. However, despite the scientific community's admittance of T.M.'s significant contribution to the achievement of health unit goals, some researchers argue that the pool of talented staff is in short supply, and competition for talent in health care will be further intensified [26]. The imbalance between demand and talent supply will be balanced if health agencies decide to make strategic investments in programming, training, and Management (attracting, developing, and maintaining) their workforce [27].

The purpose of this systematic literature review was the exploration of T.M. factors in healthcare organizations in a general pattern and the investigation of the results from the implementation of the T.M. strategies and methods.

## 2. Materials and Methods

### 2.1. Search Strategy

Systematic literature review and qualitative content analysis were used to conduct the research. A systematic literature review aims to guide researchers to materialize literature reviews in a complete, systematic, objective, reliable, and reproducible way. The various methodologies in conducting a systematic literature review have widely been described in the international literature [28,29]. The present systematic literature review methods were materialized on the directives of the Centre for Reviews and Dissemination of York University [30–32]. The Centre for Reviews and Dissemination specializes in evidence synthesis, assembling and evaluating data from multiple research studies to generate robust evidence to inform health policy and practice. The first step was to determine the purpose and goal of our research, the article selection criteria, the strategic search procedure, and the justification of any modifications. The second step was to define the search questions. The third step was to implement the systematic search of the databases.

The fourth step was to evaluate the quality and the reliability of the articles. The fifth step was the presentation and discussion of the findings in a systematic way.

The purpose and goal of the search were to undertake a systematic literature review to identify these factors related to talent management practices suitable for healthcare organizations.

### 2.2. Selection Process

All authors participated in the process of selecting eligible items for inclusion. Author K.D.M. performed the searches and undertook the initial screening of titles and abstracts against inclusion criteria. Authors M.A.T. and D.L. independently participated in the second screening of titles and abstracts. K.D.M. and M.A.T. then undertook the read-through of selected full-text articles. If there were any questions of inclusion eligibility, the fourth independent reviewer assessed the full-text item suitability.

### 2.3. Strategic Search

Electronic databases were searched for systematic literature review. These databases were PubMed, CINAHL, Cochrane Database of Systematic Reviews, Health Source/Nursing Academic Edition. Their content comprises publications relevant to health, and we followed the directives included in published articles regarding the method of literature research [33,34]. The indexing words used were: Talent Management, Talent Management Healthcare. The search period was between 1 January 2010 and 31 September 2020, as T.M. constitutes a new scientific area.

### 2.4. Selection Criteria

Articles were eligible for inclusion if they were original peer-reviewed research studies written in the English language, using quantitative, qualitative, or mixed methodology. They were investigating T.M. factors in healthcare organizations and the benefits, view-attitudes from implementing the T.M. strategies to health. Observational studies, studies without a control group, protocols, commentaries, and conference proceedings were excluded from the above search.

### 2.5. Types of Participants

Professionals: For this review, healthcare professionals have included professionals in healthcare units irrespective of their specialty.

### 2.6. Quality and Reliability Assessment of the Articles of the Sample

2.6.1. Quality Assessment

We assessed the articles' quality in the systematic literature review using Hawker evaluation methods to undertake a systematic review of disparate material [35]. The device comprises a control group of 10 points (abstract and title, introduction and goals, methodology and data, sample taking, data analysis, ethics, partiality, results, generalisability and reproducibility, consequences, usefulness: how significant these findings are in politics and practice). The tool's 10 points are assessed using four grades (Good = 4, Average = 3, Bad = 2, Very poor = 1). The lowest point average an article can receive is 10 = very poor quality (min = 10), the maximum is 40 = good quality (max = 40).

2.6.2. Reliability

The authors independently assessed the quality of the manuscripts. If there were any arguments or assessment discrepancies, these would be resolved by a third independent reviewer.

## 3. Results

### 3.1. Selecting Relevant Articles

A total of 684 papers were identified by the search strategy, leaving 89 papers after duplicate removal. We excluded 410 articles after reading the summary because they were general articles about the T.M. After screening the titles and abstracts of the articles, we removed 167 articles that did not meet the criteria set. Finally, we selected 18 studies for full-text reading, which met the eligibility criteria. After that, we identified another six citations who met the inclusion criteria and eventually included them in the systematic review (see Figure 1 for the Preferred Reporting Item for Systematic Reviews and Meta-Analyses (PRISMA) flow diagram) [36].

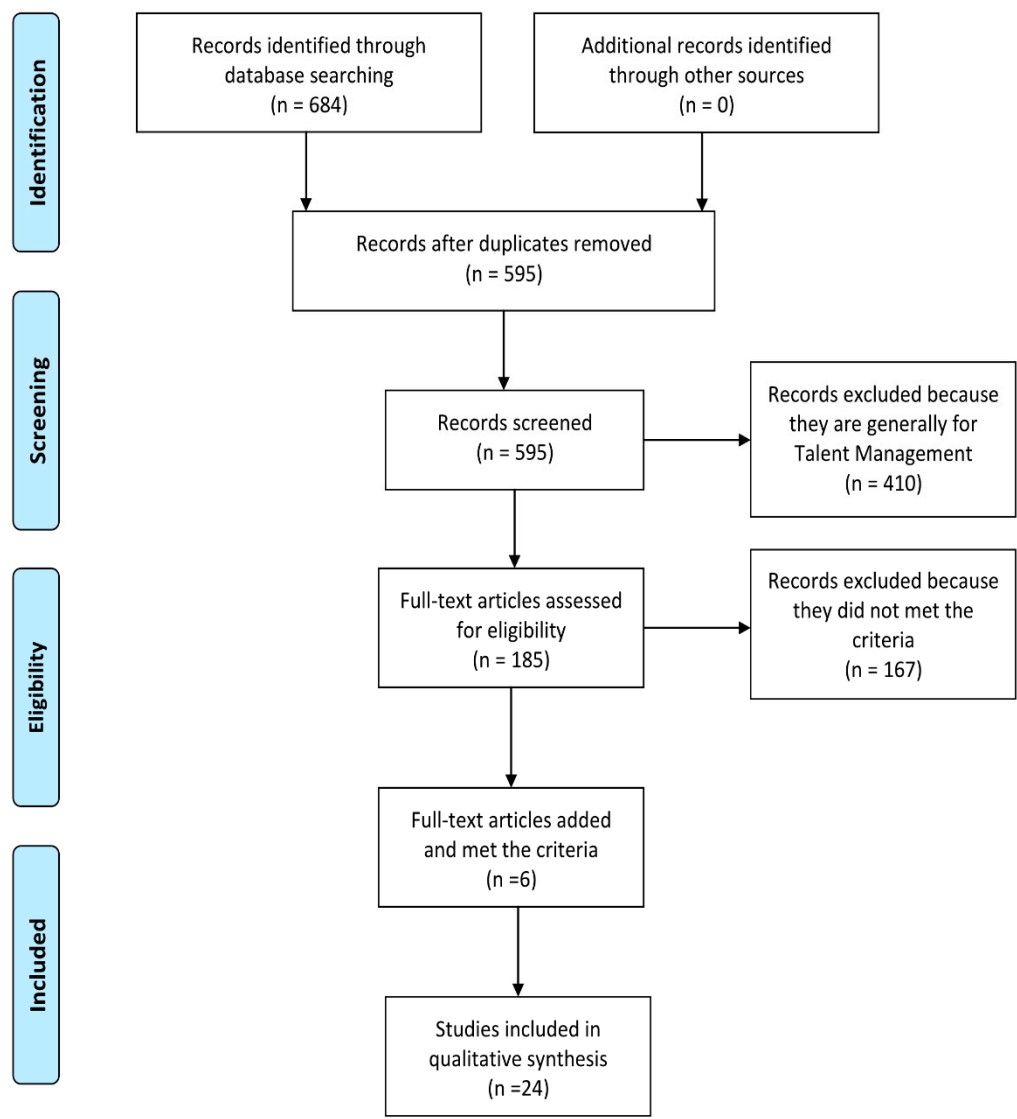

**Figure 1.** Flowchart of the article selection process.

### 3.2. Main Points of the Systematic Review

A standardized data deduction card, based on the needs of the research, was used. The following information was deduced and recorded from the analysis: authors, publication date, sample size, the methodology used, country, essential points of research.

As it becomes apparent from Table A1 in Appendix A, the research understudy's total sample was $n = 5159$ participants. From the total number of studies, the majority, 17

(70.83%) of studies were quantitative, in 3 (12.05%) qualitative methodology was used, and in the remaining 6 (16.6%) a mixed one.

The countries where the studies came from in the sample's research were 3 in Iran, 3 in Egypt, 5 in Jordan, 3 in Kenya, and one in Georgia, Australia, England, Pakistan, Libya, Poland, Slovakia, Africa, Malaysia, and New Zealand. The total number of publications per year of publication is presented (1:2011, 0:2012, 1:2013, 2:2014, 5:2015, 5:2016, 2:2017, 3:2018, 4:2019, 1:2020), following an upward trend. Finally, the systematic literature review research was in English, referred to original research studies, published in international journals after being peer-reviewed, and answered the research questions.

### 3.3. Reliability

No article received a failing grade due to low quality. The quality of the articles included in the systematic review is presented in Table 1.

**Table 1.** Quality assessment of the articles of the systematic review.

| S/N | Authors | Year | Sample | Journal | Total Score |
|---|---|---|---|---|---|
| 1 | Taha, V.A., Gajdzik, T., Zaid, J.A. [37] | 2015 | 154 | European Scientific Journal | 29 |
| 2. | Ingram, T., Glod, W. [38] | 2016 | 60 | Procedia Economics and Finance | 34 |
| 3. | Irtaimeh, J.H. [39] | 2016 | 983 | International Journal of Management | 38 |
| 4. | Nafei, W.A. [40] | 2015 | 285 | American International Journal of Social Science | 33 |
| 5. | Taie, S.M. [41] | 2015 | 119 | American Research Journal of Nursing | 36 |
| 6. | Karemu, G.K., Kachori, D., Josee, V.M., Okibo, W. [42] | 2014 | 40 | European Journal of Business and Management | 30 |
| 7. | Nel, P., Rodriques, W. [43] | 2015 | 115 | NZJHRM | 29 |
| 8. | Funk, L.M., Conley, D.M., Berry, W.R., Gawande, A.A.A. [44] | 2013 | 54 | World Journal of Surgery | 30 |
| 9. | Kheirkhah, M., Akbarpouran, V., Haqhani, H. [45] | 2016 | 177 | Journal of Client-Centered Nursing Care | 30 |
| 10. | Irtaimeh, H.J., Al-Azzam, J.F., Khaddam, A.A. [46] | 2016 | 120 | Journal of Entrepreneurship & Organization Management | 38 |
| 11. | Elarabi, M.H., Johari, F. [47] | 2014 | 179 | Asian Social Science | 27 |
| 12. | Hashemzaee, A., Ghasemi, M. [48] | 2017 | 209 | Science Arena Publications Specialty Journal of Psychology and Management | 33 |
| 13. | Abedi, G., Ahmadi, A., Asi, M.I. [49] | 2011 | 436 | Middle-East Journal of Scientific Research | 33 |
| 14. | Subramaniam, A., Silong, A.D., Uli, J., Ismail, A.I. [50] | 2015 | 335 | BMC Medical Education | 34 |
| 15. | Ngaira, P., Benard, O. [51] | 2016 | 102 | International Journal of Business and Management Invention | 28 |
| 16. | Cabral, A., Oram, C., Allum, S. [52] | 2019 | 81 | Journal of Nursing Management | 30 |
| 17. | Bibi, M. [53] | 2019 | 364 | SEISENSE Journal of Management | 28 |
| 18. | Mwanzi, J., Wamitu, S., Kiama, M. [54] | 2017 | 454 | IOSR-JBM | 28 |
| 19. | Leggat, S.G., Liang, Z., Hward, P.F. [55] | 2020 | 64 | Australian Health Review | 34 |
| 20. | Smith, A.D., Arnold, W.L., Krupinski, E.A., Powell, C., Meltzer, C.C. [56] | 2019 | 100 | Journal of American College Radiology | 30 |
| 21. | El Dashan, A.E.M., Keshk, L.M., Dorgham, L.S. [57] | 2018 | 273 | International Journal of Nursing | 34 |
| 22. | Irtaimeh, H.J., AL-Azzam, J.F., Khaddam, A.A. [58] | 2019 | 135 | Management & Applied Economics Review | 36 |
| 23. | Obeidat, B., Yassin, H., Masa'deh, R. [59] | 2018 | 251 | Modern Applied Science | 35 |
| 24. | Atoom, A.D. [60] | 2018 | 77 | International Journal of Business Administration | 32 |

*3.4. Type of Participants*

The type of participants from the 24 selected studies is presented in Table 2.

**Table 2.** Type of participants.

| Paper | Participants |
|---|---|
| Taha, V.A., Gajdzik, T., Zaid, J.A. [37] | 154 Employees of a hospital, regardless of their specialty |
| Ingram, T., Glod, W. [38] | 60 Managers and executives familiar with Talent Management issues |
| Irtaimeh et al. 2016 [46] | Manager/Assistant manager = 13, Head section = 41, Administrative Staff = 66 |
| Nafei, W.A 2015 [40] | Physicians = 105<br>Nurses = 164<br>Administrative staff = 16 |
| Taie 2015 [41] | 119 Three-tier senior executives (senior = 13, middle = 31 and foremost = 65) |
| Karemu et al. 2014 [42] | Doctors = 25, Nurses = 15 |
| Nel and Rodriques 2015 [43] | 107 Staff of various specialties 8 Managers |
| Funk, L.M., Conley, D.M., Berry, W.R., Gawande, A.A. [44] | 54 Managers, medical and technical staff |
| Kheirkhah et al. 2016 [45] | 177 Midwives |
| Irtaimeh 2016 [39] | 983 Staff and patients in hospitals |
| Elarabi & Johari 2014 [47] | 179 Managers staff, medical and other hospital staff |
| Hashemzaee & Ghasemi 2017 [48] | 209 Employees of a university hospital, regardless of their specialty |
| Abedi et al. 2011 [49] | 130 Managers, 306 Hospital Nurses |
| Subramaniam et al. 2015 [50] | 335 Trainee doctors from six public hospitals |
| Ngaira & Benard 2016 [51] | Medical officers = 22<br>Senior officers = 22<br>Others = 58 |
| Cabral et al. 2019 [52] | 81 NHS directors of nursing, chief nurses, directors of quality and their deputies in southeast England |
| Bibi, M. 2019 [53] | 364 employees including paramedical & administrative staff and physicians working in different healthcare organizations in Karachi, Pakistan |
| Mwanzi et al. 2017 [54] | 241 Doctors, 67 nurses, 21 pharmacists, 27 lab technicians, 9 radiographers, 20 accountants, 11 physiotherapists, 21 secretaries and 31 cleaners |
| Leggat et al. 2020 [55] | CEO = 19, Level II = 23, Level III = 11, Board member = 11 |
| Smith et al. 2019 [56] | 100 Academic Executives |
| Dahshan et al. 2018 [57] | Staff Nurse = 217<br>Nurse Manager = 56 |
| Irtaimeh et al. 2019 [58] | Manager/Assistant manager 13<br>Head Section 41<br>Staff 66 |
| Obeidat et al. 2018 [59] | 251 Employees of the independent specialty of Private Hospital |
| Atoom 2018 [60] | 77 Doctors of Public Hospitals |
| Total sample | *n* = 5159 |

## 4. Discussion

Qualitative content analysis constitutes a systematic method used to analyze data compiled through the qualitative, quantitative, or mixed methodology. It focuses on how specific and predetermined issues are approached in an article and on the frequency they appear. The particular method was widely used in many sciences, such as psychology, sociology, medicine, communication science, and many more [61].

Qualitative content analysis is used by researchers when adequate previous knowledge about a particular issue is absent. The ultimate purpose of qualitative content analysis is to draw out gaps in the bibliography, explore them with new tools and verify previous theories and hypotheses in such a way as to produce modern knowledge [62].

The purpose of qualitative content analysis is to describe the characteristics of an article's context, examining who says what, to whom, and with what result. The qualitative content analysis must be perceived and understood as a data analysis technique within a framework of a procedure guided by rules and linked to standard qualitative research models [63]. From time to time, the method of qualitative content analysis has received several criticisms of its simplicity. Nonetheless, we know that if the issue studied comprises sensitive data such as that on the field of health, the flexibility of the method is determinant in arriving at the appropriate conclusions [64].

The present qualitative content analysis constitutes a secondary study aiming to compile and record already existing T.M. data in healthcare organizations. In this way, we will identify gaps in the literature on Talent Management factors and the results from T.M. methods' implementation to health while verifying or discarding previous theories and hypotheses for contemporary knowledge to be produced.

The results from the systematic literature review of the 24 studies were grouped, and the following derived categories:

### 4.1. Programming

Programming constitutes the primary and possibly the most challenging step for an organization to implement T.M. strategies. Whatever conscious change is to be materialized, it should first be understood to be readily acceptable and useful. Programming is the in-depth comprehension of the organization's business goals and its competitive environment. It is a combination of comprehending and predicting demand as well as funds availability. Nafei, W.A, among others, maintained that T.M. programming is one of the most significant T.M. factors [40]. At the same conclusion arrived another research, stressing the significance of both in-depth comprehensions of hospital goals and broad exploration of available human resources [49].

### 4.2. Attraction

Talented personnel attraction, whether from the internal or the external organization environment, was acknowledged as one of the most significant T.M. dimensions, which is also proven by the total number of studies supporting its significance. More specifically, several studies [43,45] argued that attraction constitutes a significant factor for T.M.

Another study found that to attract talented personnel either from the external or the internal environment, the organization should possess the appropriate T.M. culture to attract talents [33]. Finally, Obeidat, in addition to the positive correlation between the TM factors (attraction, growth, retention) investigated, concluded that each factor individually could positively affect organizational effectiveness (satisfaction, organizational participation) [59].

### 4.3. Development

Talent development is perceived as a continuous process aiming to either develop the abilities and skills of the existing organization personnel for new talents to arise or develop the acquired talented employees' abilities to assume leading positions. The development factor's significance to T.M. was verified by four studies [42,45]. Furthermore, organizations should implement broadened T.M. strategies and equally broadly examine the former's individual goals and needs. Moreover, some researchers stressed the importance of timely and organized support of talented candidates combined with their internship and the more personalized support of their role to develop their diverse skills [52]. On the other hand, Leggat et al. concluded that T.M.'s global strategic approach across the public sector offers more significant benefits than T.M. at the individual level [55].

### 4.4. Preservation

Besides attracting and developing talents, preservation also constitutes a difficult part of T.M. in organizations since several factors influence it. Two studies concluded, among others, that conservation is the most significant of the T.M. dimensions [40,49,58]. Furthermore, another study concluded that professional development opportunities, work climate and education opportunities, available training and development levels and remuneration, and benefits attraction positively impacted preserving doctors and nurses who served as samples to their research [42]. Moreover, Atoom proposed to the hospitals' administrations of his study the development of appropriate T.M. strategies and providing all the necessary tools to the medical staff to retain the talents and increase their efficiency [60].

Studies also maintained that T.M. strategies, including preservation, directly influence personnel performance and satisfy patients [47]. In addition to patient satisfaction, another study found a positive correlation between retention and employees' satisfaction of independent specialty that was the sample of their research and organizational participation [59].

Finally, one study discovered a positive correlation between employee preservation strategies and hospital organizational performance [51]. In the same direction were the researchers' findings, who found a strong correlation between all T.M. factors (attracting, retaining, developing, motivating) and organizational performance [57].

### 4.5. Performance Assessment

Through personnel performance assessment, organizations succeed in reinforcing their performance and improving the quality of services provided. The personnel performance assessment constitutes a significant T.M. factor, provided that assessment results coincide with employee position improvement [37]. The conclusion from another research moves in a similar direction. The development and maintenance of a staff competency assessment center in the public sector are essential for the T.M. [55].

Nafei, W.A coined the idea according to which provided services quality will increase if, within T.M.'s framework, regular personnel self-assessment is implemented [55]. In some studies, it was also argued that a lot of difficulty in T.M. arises, among other things, from the non-implementation of continuous self-assessment programs [40]. Moreover, other authors also suggested that for personnel performance and satisfaction improvement, constant feedback between managers and the employees aims to improve work conditions and personnel performance [47]. Finally, researchers concluded that through Talent Management, healthcare providers could increase both employees' and their organization's performance by continually evaluating and reviewing their strategies [53].

### 4.6. Work Climate

Work conditions constitute a significant factor for Human Resources Management dimensions. When an organization's produced outcome is sensitive to health, the work climate for the employees, whose conditions are characterized as "work intensity," definitely constitutes a determinant factor. Two studies [42,43] respectively proved that the work climate is among the significant T.M. factors. Both types of research concluded that T.M. strategies and work climate might influence preserving talents in organizations. Moreover, by improving employees' working environment and retaining the skill, you create the right conditions for the exogenous motivation of talent and work performance [54]. Additionally, some authors support that T.M. strategies, such as work climate, positively influence healthcare efficiency [43].

### 4.7. Culture

"A culture is a group of values, symbols and rituals shared between members of a particular organization, which describe the actions undertaken to resolve new situations related to internal management and also those related to clients and suppliers" [23]. Therefore, the culture of an organization regarding specific issues proves how these are

confronted in an organization. From this aspect, various viewpoints have been introduced related to the practice of talent management, stating that a careful approach to talent management centered on providing all candidates with equal opportunities is recommended [65]. In another research, the authors believe that T.M. should strive to support human resources, even the potentially employed, as a total [38].

One study maintained that organizational culture, among other things, constitutes a significant T.M. factor and is strongly and positively correlated with the quality of services provided [37]. Another study found a strong positive correlation between organizational culture and personnel preservation [63].

### 4.8. Succession Planning

Reviewing many research studies in international literature, we concluded that talent succession planning in an organization emerges as a significant T.M. factor. Also, there is a need for talented personnel tanks due to competition for healthcare talents.

Research on University Hospital employees, irrespective of specialty, argued that talent succession planning constitutes a significant T.M. factor [48]. Succession planning is an essential factor in Talent Management. It was also mentioned in another research study, where the researchers concluded that T.M. (attracting, developing, retaining, succession planning) positively affects the quality of services and the quality of its beneficiaries' satisfaction. T.M. and quality also positively correlate with independent variables with satisfaction [58]. In another survey, T.M. strategies aim to explore talents among the entire personnel, regardless of the employment level, even in the case of those potentially employed [46]. Finally, some researchers support that both the standardization of the networks of the candidate talents and the holistic team approach in their training is essential for the acquisition of the necessary supplies required by their position [52].

### 4.9. Leadership

Leaders of organizations are the ones who motivate the entire workforce of the organization to the achievement of its vision and strategic planning. It follows that the leaders are those who should promote and support the implementation of T.M. strategies.

Leadership as a necessary T.M. factor positively influences the manager's selection. The manager's practice in T.M. matters should be a continuous process for organizational success to remain high [41,49].

The various forms of leadership, which trainee doctors' supervisors exercise, are positively correlated with T.M. More specifically, in one study, the authors believed that doctors' supervisors could develop the appropriate work climate for the trainee to create their specific talents [50]. A more specialized approach concluded that supervision on strategies exercised by the manager positively influenced organizational performance [51]. Another conclusion stated that leadership development is necessary for organizations that depend on talents such as academic centers to survive in a competitive environment with increasingly limited resources [56].

In Table 3, and Table A1. Appendix A, we present the summary of the 9 Talent Management factors derived from the qualitative content analysis.

**Table 3.** The summary of T.M. factors resulted from the qualitative content analysis.

| Number | T.M. Factors | References |
|:---:|:---:|:---:|
| 1 | Programming | Nafei, W.A 2015 [40], Abedi et al. 2001 [49] |
| 2 | Attraction | Nel & Rodriques 2015 [43], Kheirkhah et al. 2016 [45], Irtaimeh et al. 2016 [46], Hashemzaee et al. 2017 [48], Obeidat et al. (2019) [59], Atoom (2018) [60] |
| 3 | Development | Karemu et al. 2014 [42], Taha et al. 2015 [37], Kheirkhah et al. 2016 [45], Ingram & Glod 2016 [38], Dahshan et al. (2018) [57], Obeidat et al. (2019) [59], Atoom (2018) [60], Cabral et al. (2019) [52], Leggat et al. (2020) [55] |
| 4 | Preservation | Hashemzaee et al. 2017 [48], Irtaimeh 2016 [39], Karemu et al. 2014 [42], Elarabi & Johari 2014 [47], Ngaira & Benard 2016 [51], Atoom (2018) [60], Obeidat et al. (2019) [59], Dahshan et al. (2018) [57] |
| 5 | Performance assessment | Taha et al. 2015 [38], Nafei, W.A 2015 [40], Elarabi & Johari 2014 [47], Nel & Rodriques 2015 [43], Leggat et al. (2020) [55], Bibi (2019) [53] |
| 6 | Work climate | Nel & Rodriques 2015 [43], Karemu et al. 2014 [42], Mwanzi et al. (2017) [45] |
| 7 | Culture | Ingram & Glod 2016 [38], Irtaimeh et al. 2016 [46], Nel & Rodriques 2015 [43], Nafei, W.A 2015 [40] |
| 8 | Succession planning | Hashemzaee et al. 2017 [48], Irtaimeh et al. 2016 [46], Irtaimeh et al. (2019) [58] Cabral et al. (2018) [52] |
| 9 | Leadership | Abedi et al. 2011 [49], Ngaira & Benard 2016 [51], Subramaniam et al. 2015 [7], Taie 2015 [41], Smith et al. (2019) [56] |

## 5. Conclusions

The present systematic literature review explored T.M. factors in healthcare services organizations and T.M. strategies and methods. The systematic literature review produced 24 articles appropriate for our research, the quality, and appropriateness for further analysis with the corresponding tool. Twenty out of twenty-four research studies were published within the last five-year period (2015–2020) and came from 10 different countries. We used qualitative content analysis for the analysis of the results [63].

The systematic literature review results support previous studies' findings regarding significant T.M. factors in healthcare services organizations such as attraction, development, preservation, succession, education, official assessment, programming, leadership,

remuneration, and organizational culture. Preservation is of determinant significance to talents [66,67]. It seems that healthcare organization's strategies positively influence many factors such as the quality of services, the satisfaction of patients and personnel, the efficiency of services provided, and organizational commitment. Through T.M. strategies' implementation, organizations may improve the efficiency and effectiveness of their services, their patients and personnel satisfaction, and can acquire employees of high loyalty to the organization's values and goals [10]. The benefits for both the organizations and the personnel and the positive correlation of T.M. with the efficiency of services provided are consistent with two kinds of research [65,68]. In particular, some studies have supported the positive correlation between talent management and the organization's performance [3,60,69–71].

Moreover, leadership as a basic HRM and modern administration models factors constitutes a significant T.M. factor; in other words, leadership can influence T.M. strategies' implementation at whatever level is exercised. Effective leadership in a hospital is also an essential strategy for achieving high performance [22]. Also, to emphasize T.M. leadership's significance, other researchers maintained that human resources managers must locate structure and start T.M. systems and efforts developing organizational commitment to T.M. [1].

Employees in healthcare organizations that made up the sample of the systematic review support in their majority that T.M. ought to maintain a broader approach and consider personnel personal goals and needs. Also, talent preservation is a significant variable for the employer's trademark [13]. A different study also argued that maintaining a skilled workforce is essential in high-performance hospitals [26]. Therefore, they recommended that, for goal achievement, incomplete T.M. systems, communication between employees and human resources managers be effected in predetermined time intervals aiming at their constant update on current matters of interest to personnel, since each employee's personal needs differ and continuously change. Finally, most of the sample brought to the surface the need for a T.M. strategies dynamic approach with continuous feedback for all personnel levels. The afore finding is aligned with another research. Enterprises should design an assessment tool that will monitor all factors influencing T.M. and update T.M. strategies regarding continuous changes [45,72].

**Author Contributions:** Conceptualization, M.A.T. and K.D.M.; methodology, M.A.T., K.D.M. and D.L.; software, D.L.; writing—original draft preparation, K.D.M. and M.A.T.; writing—review and editing, M.A.T.; supervision, M.A.T. All authors have read and agreed to the published version of the manuscript.

**Funding:** This research received no external funding.

**Institutional Review Board Statement:** Not applicable.

**Informed Consent Statement:** Not applicable.

**Data Availability Statement:** Data is contained in the article.

**Conflicts of Interest:** The authors declare no conflict of interest.

# Appendix A

Table A1. The main points of the articles of the systematic review.

| S/N | Article | Sample | Method | Country | Basic Points |
|---|---|---|---|---|---|
| 1 | Taha, V.A., Gajdzik, T., Zaid, J.A. [37] | 154 Employees of a hospital, regardless of their specialty | Quantitative | Slovakia | The size of the organization has an impact on the T.M. process implementation. The education and training for the continuous development of personnel abilities is an essential factor. The development of personnel (placing the right employee at the right post), and official performance assessment regularly are significant T.M. factors. Employers in health care pay little attention to the establishment of policies encouraging career development and progress potential. The implementation of regular official performance assessment has been underrated. |
| 2. | Ingram, T., Glod, W. [38] | 60 Managers and executives familiar with Talent Management issues | Qualitative Semi-structure interviews | Poland | For the employees to fully develop their abilities, organizations ought to implement broader T.M. strategies. Financial incentives and educational activities alone cannot maintain employee satisfaction at a high level. For the personnel to be preserved, organizations should more broadly examine the former's individual goals and needs. A more careful T.M. approach is recommended, centered on providing equal opportunities to all candidates and not only to top brass medical personnel. |
| 3. | Irtaimeh et al. 2016 [46] | 120 Employees of a hospital, regardless of their specialty | Quantitative Questionnaire | Jordan | T.M. strategies are strongly correlated as much with quality as with beneficiary (patient–client) satisfaction. Quality has a powerful influence on beneficiary satisfaction. Researchers believe that T.M. should strive to support human resources as a total, even the potentially employed. |
| 4. | Nafei, W.A 2015 [40] | 285 Doctors, Nurses and Support Staff | Quantitative Questionnaire | Egypt | Programming, recruitment performance measures, reinforcement, involvement, and organizational culture constitute significant T.M. factors and are strongly and positively correlated with the quality of services provided. In T.M.'s framework, the researcher believes that innovative self-assessment actions should be implemented. |

**Table A1.** *Cont.*

| S/N | Article | Sample | Method | Country | Basic Points |
|---|---|---|---|---|---|
| 5 | Taie 2015 [41] | 119 Three-tier senior executives (senior, middle and foremost) | Quantitative Questionnaire | Egypt | There has been a statistically significant difference between T.M.'s administrative nursing personnel knowledge before and after their sensitization interviews. There has been a strong positive correlation between T.M. elements and organizational success results. The researcher proposes long-lasting training of the managers in T.M. skills. |
| 6. | Karemu et al. 2014 [42] | 40 Doctors and Nurses | Mixed Questionnaire, semi-structured interviews | Kenya | T.M. strategies positively influence the preservation of medical and nursing personnel. They reinforce the provision of better quality services. Professional development opportunities, work climate, educational opportunities, available training, and development levels, and attractive remuneration and benefits have positively influenced doctors' and nurses' preservation. |
| 7. | Nel and Rodriques 2015 [43] | 107 Staff of various specialties 8 Managers | Mixed Questionnaire, semi-structured interviews | New Zealand | T.M. factors such as attraction, education, career prospects, work conditions, work culture, and trust are strongly and positively correlated with personnel preservation. T.M. positively influences the efficiency of health care. The difficulty arises from T.M. strategies' implementation point to low personal work satisfaction, vague alternatives to career development, education and training, self-evaluation program implementation, and remuneration targeting a balance between personal life and work. |
| 8. | Funk, L.M., Conley, D.M., Berry, W.R., Gawande, A.A. [44] | 54 Managers, medical and technical staff | Qualitative Semi-structured interviews | Sub-Saharan Africa | Goal determination, operations management, T.M., quality pursuit, and financial supervision can increase productivity and surgical interventions' effectiveness and bear positive results. |
| 9 | Kheirkhah et al. 2016 [45] | 177 Midwives | Quantitative Questionnaire | Iran | Attraction, preservation, and development of talents are strongly positively correlated with organizational commitment. An increase in commitment will lead to a reduced work cycle, increased personnel performance, and improved service quality. To attract talents, the organization should possess the necessary culture. |

Table A1. *Cont.*

| S/N | Article | Sample | Method | Country | Basic Points |
|---|---|---|---|---|---|
| 10. | Irtaimeh 2016 [39] | 983 Staff and patients in hospitals | Quantitative Questionnaire | Jordan | Talent attraction, development, preservation, and succession strategies are the most significant dimensions of T.M. The implementation of T.M. methods impacts the quality of services provided as much as the satisfaction of patients. |
| 11. | Elarabi & Johari 2014 [47] | 179 Managers staff, medical and other hospital staff | Quantitative Questionnaire | Libya | The strong positive correlation between Human Resources Management practices, such as T.M., and the quality of health care has resulted in improved personnel performance and patient satisfaction. |
| 12. | Hashemzaee & Ghasemi 2017 [48] | 209 Employees of a university hospital, regardless of their specialty | Quantitative Questionnaire | Iran | Talent attraction, recognition, development, preservation, proper use, and creation constitute significant factors. T.M. and motivation to work are positively correlated with conflict management. The presence of talents in organizations increases personnel creativity and efficiency. |
| 13. | Abedi et al. 2011 [49] | 130 Managers 306 Hospital Nurses | Quantitative Questionnaire | Iran | T.M. factors such as planning, development, change, effective communication, leadership, teamwork, productivity, goal achievement, and personnel management influence the manager's selection. |
| 14. | Subramaniam et al. 2015 [50] | 335 Trainee doctors from six public hospitals | Quantitative Questionnaire | Malaysia | Supervisory forms of trainee doctors, such as coaching supervision and mentoring supervision, are positively correlated with talent development. There is no significant relationship between abusive supervision and talent development. Besides, trainee doctor supervisors can develop academic units to meet trainee doctors' needs and create an environment encouraging them to apply their skills and develop their talents. |
| 15. | Ngaira & Benard 2016 [51] | 102 Employees of the Ministry of Health | Mixed Questionnaire, semi-structured interviews | Kenya | The relation between recruitment strategies and organizational performance of public hospitals in the Mombasa region was found to be positive and significant. A significant correlation was found between all employee preservation strategies and hospital organizational performance to be more specific. It was also proven that supervision strategies as a primary specialized personnel preservation factor also influences organizational performance. |

**Table A1.** *Cont.*

| S/N | Article | Sample | Method | Country | Basic Points |
|---|---|---|---|---|---|
| 16. | Cabral et al. 2019 [52] | 81 Nursing executives and their deputies | Qualitative, Semi-structure interviews | England | The development and identification of talents require timely and organized information, with practical experience and personalized support. Standardization of networks for potential candidates is considered critical. Finally, a holistic team approach to training and mentoring talent candidates is essential. |
| 17. | Bibi, M. 2019 [53] | 364 Healthcare employees | Quantitative Questionnaire | Pakistan | Talent Management Factors such as attraction and selection, guidance and support, compensation have a strong positive correlation with employee performance. By reviewing T.M. practices, healthcare organizations can increase the productivity of both employees and their organizations. |
| 18. | Mwanzi et al. 2017 [54] | 454 Healthcare employees regardless of specialty | Quantitative Questionnaire | Kenya | A positive correlation is found between T.M. and organizational development. Talent development and the work environment influence the exogenous motivation of talents and the improvement of performance. It is proposed to meet employees' needs by organizations and provide favorable working conditions to increase efficiency, effectiveness, productivity, and work commitment. |
| 19. | Leggat et al. 2020 [55] | 64 Senior, Medium, and Senior Managers | Mixed Questionnaire, semi-structured interviews | Australia | The development and maintenance of a staff appraisal center is a critical element of T.M. Implementing a holistic strategic approach to the public healthcare sector is vital to attracting talent from an external pool. T.M. strategies (attracting, developing, retaining, assessing competencies) contribute positively to reducing healthcare costs. |
| 20. | Smith et al. 2019 [56] | 100 Academic Executives | Quantitative Questionnaire | Georgia | The development of leaders at all levels of organizations is an essential factor of I.T., primarily in organizations whose performance depends directly on talents' performance. |
| 21. | Dahshan et al. 2018 [57] | 273 Nursing staff | Quantitative Questionnaire | Egypt | There is a strong positive correlation between T.M. (attracting, retaining, motivating, and developing) and organizational performance. A positive correlation was found between the factors separately and the organizational performance. |

**Table A1.** *Cont.*

| S/N | Article | Sample | Method | Country | Basic Points |
|---|---|---|---|---|---|
| 22 | Irtaimeh et al. 2019 [58] | 135 Employees of an independent specialty of the Pediatric Hospital | Quantitative Questionnaire | Jordan | The T.M. and the quality of the services provided as independent variables affect the satisfaction of their beneficiaries. Also, from the results, you can conclude that T.M. affects the quality and quality of the beneficiaries' satisfaction. |
| 23. | Obeidat et al. 2018 [59] | 251 Employees of the independent specialty of Private Hospital | Quantitative Questionnaire | Jordan | T.M. factors (attraction, growth, retention) have a strong positive correlation with organizational efficiency (job satisfaction, organizational involvement) both as a whole and as individual factors. Also, the aspects of organizational efficiency were significantly correlated with each other. |
| 24. | Atoom 2018 [60] | 77 Doctors of Public Hospitals | Quantitative Questionnaire | Jordan | The teaching hospitals' administrations must develop systems and programs to attract, select, develop, evaluate talent, and provide all the necessary tools to retain medical talent. T.M. affects all dimensions of the sample physicians' performance (presentation of morning report, morning round, examination and acceptance of patients on the night and night shifts, and coordination with other hospital departments for diagnostic procedures). |
| Total sample | | | | | 5159 |

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
