# Peer review of "Talent Management in Healthcare: A Systematic Qualitative Review"

_sustainability, doi:10.3390/su13084469_

Round 1
Reviewer 1 Report
This article is interesting because they carry out analysis work on existing documents on talent management in healthcare. It is a second-level investigation carried out from existing relevant articles. The article is well structured and clear. Here are some recommendations to improve the article:
- In line 85, you indicate that you want to do future research. What will this search bring you in future research? What kind of future research do you plan to do?
- You introduce the concept "meta-analysis" (line 90). A meta-analysis is different from a systematic review. This should be clearer as this paragraph can be misleading. Clearly differentiate these two concepts. Please state: Why have you opted for a systematic review? Why haven't they done a meta-analysis?
- In line 119, you indicate that do you use different kinds of professionals. They can indicate the type of professionals that are part of the articles. (At least some examples)
- In line 123, it indicates that you use a "widely used tool". You can specify the name of the tool. What is the name of the tool?
- Indicate the limitations of a study of this type.
- On line 368, indicate more specifically the "gaps in the literature" found in this study (as you indicated on line 192).
- Have you been able to confirm any hypotheses with your study?
Author Response
Dear Reviewer
Thank you for your constructive comments. We have thoroughly revised the manuscript in accordance with your suggestions and consider it greatly improved as a result. Below, we respond to your comments below in a point-by-point fashion.
***********************************************************
This article is interesting because they carry out analysis work on existing documents on talent management in healthcare. It is a second-level investigation carried out from existing relevant articles. The article is well structured and clear. Here are some recommendations to improve the article:
Answer: Thank you for the above comment and the positive words.
************************************************************
In line 85, you indicate that you want to do future research. What will this search bring you in future research? What kind of future research do you plan to do?
Answer: Thank you for the above comment. Having reviewed the papers, we notice that there few papers in Talent Management in the Healthcare Sector. We have suggestions for future research in the discussion and conclusion.
****************************************************************
You introduce the concept "meta-analysis" (line 90). A meta-analysis is different from a systematic review. This should be clearer as this paragraph can be misleading. Clearly differentiate these two concepts. Please state: Why have you opted for a systematic review? Why haven't they done a meta-analysis?
Answer: Thank you for the above comment. Clearly, the paper deals with systematic review only. This is not a meta-analysis paper, and we have removed lines and comments related to meta-analysis. Meta-analysis refers to the statistical analysis of the data from independent primary studies focused on the same question, which aims to generate a quantitative estimate of the studies based on the evidence from each paper.
In our case, the selected paper does not have common evidence-based, not a certain quantitative parameter that is needed for the implementation of Meta-analysis. More specifically, in this review, the heterogeneity in relation to design, objectives, and results did not lend themselves to meta-analysis.
**************************************************************
In line 119, you indicate that do you use different kinds of professionals. They can indicate the type of professionals that are part of the articles. (At least some examples)
Answer: Thank you for the above comment. The professionals are medical, nursing, paramedical, administrative staff.
***************************************************************
In line 123, it indicates that you use a "widely used tool." You can specify the name of the tool. What is the name of the tool?
Answer: Thank you for the above comment. The tool does not have a specific name, and we call Hawker et al. tool and is essentially a checklist for evaluating the methodological quality of articles.
Hawker S, Payne S, Kerr C, Hardey M, Powell J. Appraising the evidence: reviewing disparate data systematically. Qual Health Res. 2002 Nov;12(9):1284-99. doi: 10.1177/1049732302238251. PMID: 12448672.
*****************************************************************
Indicate the limitations of a study of this type.
Answer: Thank you for the above comment. As indicated by Hawker et al., Qualitative research into a systematic review is often difficult, in particular at the assessment stage. This is not because qualitative research lacks relevance or rigor (though clearly sometimes it lacks both) but has much to do with the way qualitative studies are reported and presented
***************************************************************
On line 368, indicate more specifically the "gaps in the literature" found in this study (as you indicated on line 192).
Have you been able to confirm any hypotheses with your study?
Answer: Thank you for the above comment. We have not confirmed any quantitative hypothesis. However, The present systematic literature review explored TM factors in healthcare services organizations in a general way and TM strategies and methods. The systematic literature review produced 24 articles appropriate for our research. Twenty out of twenty-four research studies were published within the last five-year period (2015-2020) and came from 10 different countries. We used Qualitative content analysis for the analysis of the results
None of the studies have approached in a general-universal manner. Another critical gap that arose from the systematic review of the literature concerns the type of leadership that promotes Talent Management strategies. Further investigation of important Talent Management factors and preventing failure will prove useful in improving Talent Management in health. The present study results showed that Talent Management is a very new field of Human Resource Management in healthcare with a limited number of studies. The various variations of the objectives of the methods and the results of the studies included in the systematic review make it practically difficult to make an overall conclusion about the factors influencing Talent Management in healthcare and the results from their application. However, based on the results of the review, factors such as planning, attracting, retaining, developing, evaluating performance, work for climate, culture, leadership, and succession planning are important factors in Talent Management. Finally, the benefits identified relate to the quality of services, staff productivity, and the efficiency of the organization.
Reviewer 2 Report
The topic of the article is interesting and timely. The scope of literature sources analyzed is solid.
What methodology was used to analyze the articles in depth and systemize the knowledge? It seems, that the authors did not use any more advanced methodology for literature analysis and presentation the results.
Please provide in more detail.
Line 94: “….literature review methods were materialized on the directives of the Centre for Reviews and Dissemination of York University” please provide briefly the methodology and the justification for its choice.
In case you make qualitative analysis of the articles in publications indexed in the WoS, Scopus, or ABS list, etc., I would strongly recommend providing investigation's' motivation, why do you do that, what are you looking for. What were the criteria for qualitative evaluation?
The authors insights in the discussion part is quite narrow – it should be developed to show a value added of this paper.
Other comments.
“1.1. Aim” – It is not necessary to show the aim as a separate sub-section (it is too short).
Please check correctness of citation, for instance,
Line 95: “Centre for Reviews and Dissemination of York University [12].” Does not much to 12. AlMannai, M.A.W., A.M. Arbab, and S. Darwish, The Impact of talent management strategies on enhancement of competitive 413 advantage in Bahrain post. International Journal of Core Engineering & Management, 2017. 4(6): p. 1-17., etc.
“2.2. Selection process”- Author Contributions is provided at the end of the paper.
2.5 Types of participants – please describe more precisely – how many? Justification of their expertise.
Although keywords for literature review can be predicted, they need to be provided in the methodological section.
Author Response
Dear Reviewer
Thank you for your constructive comments. We have thoroughly revised the manuscript in accordance with your suggestions and consider it greatly improved as a result. Below, we respond to your comments below in a point-by-point fashion
Comments and Suggestions for Authors
The topic of the article is interesting and timely. The scope of literature sources analyzed is solid
Answer: Thank you for the above comment.
********************************************************
.
What methodology was used to analyze the articles in depth and systemize the knowledge? It seems, that the authors did not use any more advanced methodology for literature analysis and presentation the results.
Please provide in more detail.
Answer: Thank you for the above comment. The resulting Talent Management Factors were grouped in 9 categories: Programming, Attraction, Development, Preservation, Performance Assessment, Work climate, Culture, Succession Planning, and Leadership.
We have not confirmed any quantitative hypothesis. However, The present systematic literature review explored TM factors in healthcare services organizations in a general way and TM strategies and methods. The systematic literature review produced 24 articles appropriate for our research. Twenty out of twenty-four research studies were published within the last five-year period (2015-2020) and came from 10 different countries. We used Qualitative content analysis for the analysis of the results
None of the studies have approached a healthcare provider in a general-universal manner. Another critical gap that arose from the systematic review of the literature concerns the type of leadership that promotes Talent Management strategies. Further investigation of important Talent Management factors and preventing failure will prove useful in improving Talent Management in health. The present study results showed that Talent Management is a very new field of Human Resource Management in healthcare with a limited number of studies. The various variations of the objectives of the methods and the results of the studies included in the systematic review make it practically difficult to make an overall conclusion about the factors influencing Talent Management in healthcare and the results from their application. However, based on the results of the review, factors such as planning, attracting, retaining, developing, evaluating performance, work for climate, culture, leadership, and succession planning are important factors in Talent Management. Finally, the benefits identified relate to the quality of services, staff productivity, and the efficiency of the organization.
****************************************************************
Line 94: “….literature review methods were materialized on the directives of the Centre for Reviews and Dissemination of York University” please provide briefly the methodology and the justification for its choice.
Answer: Thank you for the above comment.
*****************************************************************
In case you make qualitative analysis of the articles in publications indexed in the WoS, Scopus, or ABS list, etc., I would strongly recommend providing investigation's' motivation, why do you do that, what are you looking for. What were the criteria for qualitative evaluation?
Answer: Thank you for the above comment.
- As suggested by the title of the paper, “Talent Management in Healthcare: a systematic qualitative review,” we wanted to evaluate papers that show the way scientists from medical disciplines how they understand Talent Management in healthcare
- Search terms were ("Talent Management") and "Healthcare" )
- From the 24 papers in Table 1, most papers are not indexed in Scopus or Abs list or Wos.
- Criteria investigation is to see if the selected papers which are related to Talent Management in Healthcare can be classified into groups. The results are presented in Table 2
*****************************************************************
The authors insights in the discussion part is quite narrow – it should be developed to show a value added of this paper.
Answer: Thank you for the above comment.
****************************************************************
Other comments.
“1.1. Aim” – It is not necessary to show the aim as a separate sub-section (it is too short).
Please check correctness of citation, for instance,
Answer: Thank you for the above comment. We remove the subsection.
******************************************************
Line 95: “Centre for Reviews and Dissemination of York University [12].” Does not much to 12. AlMannai, M.A.W., A.M. Arbab, and S.Darwish,
The Impact of talent management strategies on enhancement of competitive 413 advantage in Bahrain post. International Journal of Core Engineering & Management, 2017.4(6): p. 1-17., etc.
Answer: Thank you for the above comment. We have corrected the reference number.
****************************************************************
“2.2. Selection process”- Author Contributions is provided at the end of the paper.
Answer: Thank you for the above comment. In section 2.2 we provide more detail than at the end of the paper.
*****************************************************************
2.5 Types of participants – please describe more precisely – how many? Justification of their expertise.
Answer: Thank you for the above comment. The professionals are medical, nursing, paramedical, administrative staff.
****************************************************************
Although keywords for literature review can be predicted, they need to be provided in the methodological section.
Answer: Thank you for the above comment. We added comments in the methodological section related to the keywords.
Round 2
Reviewer 2 Report
Thank you for your efforts in improving the paper. However, the corrections were not substantial. Unfortunately, I have to note that the methodological part is very weak.
1)Given the authors ’responses, I regret to note that the authors did not use any more advanced methodology for literature analysis. I would suggest the authors to study some bibliometric or scientometric methods of analysis and presentation the results as well. The research methodology needs to be significantly strengthened.
2) The same comment remains”: Line 94: “.literature review methods were materialized on the directives of the Centre for Reviews and Dissemination of York University”. I double-checked the link https://www.york.ac.uk/crd/. This is the general page of the University of York (Centre for Reviews and Dissemination ) for dissemination. So, please briefly provide the methodology and the justification for its choice.
3) Types of participants – please describe more precisely – how many? Justification of their expertise. Is not enough to say, that “The professionals are medical, nursing, paramedical, administrative staff.” Justify from a methodological perspective.
Author Response
Thank you for your comments which I found very useful.
Answer to comment 1.
Thank you for your comment. I partly disagree with your statement. The purpose of our paper was to choose articles based on electronic medical databases such as Pubmed etc, because we wanted to investigate how Health professionals understand the concept of Talent management and assess their understanding. We consider the articles' quality in the systematic literature review using Hawker evaluation methods to review disparate material systematically.
Our methodology resulted in only 24 articles based on stated criteria. I am convinced that I can not apply scientometric or bibliometric techniques in this data set, because not too many citations. We revised the abstract of our paper to give a clear statement about the purpose of our research. We believe that the paper has its value due to our methodology.
Answer to comment 2
I removed that statement from the abstract.
Answer to comment 3
We added table 2 with the relevant information
Round 3
Reviewer 2 Report
Thank you for your response. Nevertheless, I have to note that the methodological part is still very weak. You made cosmetic corrections writing the authors you relied on but did not explain the research methodology in detail.
You stress that systematic literature has been done review using Hawker evaluation methods. Please provide how you implemented the three stages of critical appraisal of retrieved articles described by Hawker. The same comment remains”: Line 94: “.literature review methods were materialized on the directives of the Centre for Reviews and Dissemination of York University”. I for the third time the reference No 1 and link https://www.york.ac.uk/crd/. This is the general page of the University of Yourk (Centre for Reviews and Dissemination ) for dissemination. So, please briefly provide the methodology and the justification for its choice.
Author Response
Thank you for your valuable comments. Taking into account your comments and the editors comments we revised our paper accordingly. We improved the methodology section by adding additional information about the center for reviews and dissemination of York university.